# Regularized Modal Regression with Applications in Cognitive Impairment Prediction

**Xiaoqian Wang**[1], **Hong Chen**[1], **Weidong Cai**[2], **Dinggang Shen**[3], **Heng Huang**[1]*
[1] Department of Electrical and Computer Engineering, University of Pittsburgh, USA
[2] School of Information Technologies, University of Sydney, Australia
[3] Department of Radiology and BRIC, University of North Carolina at Chapel Hill, USA
xqwang1991@gmail.com,chenh@mail.hzau.edu.cn
tom.cai@sydney.edu.au,dinggang_shen@med.unc.edu,heng.huang@pitt.edu

## Abstract

Linear regression models have been successfully used to function estimation and model selection in high-dimensional data analysis. However, most existing methods are built on least squares with the mean square error (MSE) criterion, which are sensitive to outliers and their performance may be degraded for heavy-tailed noise. In this paper, we go beyond this criterion by investigating the regularized modal regression from a statistical learning viewpoint. A new regularized modal regression model is proposed for estimation and variable selection, which is robust to outliers, heavy-tailed noise, and skewed noise. On the theoretical side, we establish the approximation estimate for learning the conditional mode function, the sparsity analysis for variable selection, and the robustness characterization. On the application side, we applied our model to successfully improve the cognitive impairment prediction using the Alzheimer's Disease Neuroimaging Initiative (ADNI) cohort data.

## 1 Introduction

Modal regression [21, 5] has gained increasing attention recently due to its effectiveness on function estimation and robustness to outliers and heavy-tailed noise. Unlike the traditional least-square estimator pursuing the conditional mean, modal regression aims to estimate the conditional mode of output $Y$ given the input $X = x$. It is well known that the conditional modes can reveal the structure of outputs and the trends of observation, which is missed by the conditional mean [29, 4]. Thus, modal regression often achieves better performance than the traditional least square regression in practical applications.

There are some studies for modal regression with (semi-)parametric or nonparametric methods, such as [29, 28, 4, 6]. For parametric approaches, a parametric form is required for the global conditional mode function. Recent works in [29, 28] belong to this category, where the method in [28] is based on linear mode function assumption and the algorithm in [29] is associated with the local polynomial regression. For non-parametric approaches, the conditional mode is usually derived by maximizing a conditional density or a joint density. Typical work for this setting is established in [4], where a local modal regression is proposed based on kernel density estimation and theoretical analysis is provided to characterize asymptotic error bounds.

Most of the above mentioned works consider the asymptotic theory on the conditional mode function estimation. Recently, several studies on variable selection under modal regression were also conducted in [30, 27]. These approaches addressed the problem from statistical theory viewpoint (*e.g.*,

asymptotic normality) and were implemented by modified EM algorithm. Although these studies provide us good understanding for modal regression, the following problems still remain unclear in theory and applications. *Can we design new modal regression following the line of structural risk minimization? Can we provide its statistical guarantees and computing algorithm for designed model?* This paper focuses on answering the above questions.

To illustrate the effectiveness of our model, we looked into a practical problem, *i.e.*, cognitive impairment prediction via neuroimaging data. As the most common cause of dementia, Alzheimer's Disease (AD) imposes extensive and complex impact on human thinking and behavior. Accurate and automatic study of the relationship between brain structural changes and cognitive impairment plays a crucial role in early diagnosis of AD. In order to increase the diagnostic capabilities, neuroimaging provides an effective approach for clinical detection and treatment response monitoring of AD [13]. Several cognitive tests were presented to assess the individual's cognitive level, such as Mini-Mental State Examination (MMSE) [8] and Trail Making Test (TMT) [1]. With the development of these techniques, a wide range of work employed regression models to study the correlations between neuroimaging data and cognitive measures [23, 16, 26, 25, 24].

However, existing methods use mean regression models based on the least-square estimator to predict the relationship between neuroimaging features and cognitive assessment, which may fail when the noise in the data is heavy-tailed or skewed. According to the complex data collection process [13], the assumption of symmetric noise may not be guaranteed in biomedical data. Under such a circumstance, modal regression model proves to be more appropriate due to its robustness to outliers, heavy-tailed noise, and skewed noise. We applied our method to the ADNI cohort for the association study between neuroimaging features and cognitive assessment. Experimental results illustrated the effectiveness of our model. Moreover, with sparse constraints, our model found several imaging features that have been reported to be crucial to the onset and progression of AD. The replication of these results further support the validity of our model.

Our main works can be summarized as below:

1) Following the Tikhonov regularization and kernel density estimation, we develop a new Regularized Modal Regression (RMR) for estimating the conditional mode function and selecting informative variables, which can be considered as a natural extension of Lasso [22] and can be implemented efficiently by half-quadratic minimization methods.

2) Learning theory analysis is established for RMR from three aspects: approximation ability, sparsity, and robustness, which provide the theoretical foundations of the proposed approach.

3) By applying our RMR model to the ADNI cohort, we reveal interesting findings in cognitive impairment prediction of Alzheimer's disease.

## 2 Regularized Modal Regression

### 2.1 Modal regression

We consider learning problem with input space $\mathcal{X} \subset \mathbb{R}^p$ and output space $\mathcal{Y} \subset \mathbb{R}$. Let $p_{Y|X=x}$ be the conditional density of $Y \in \mathcal{Y}$ for given $X = x \in \mathcal{X}$. In the prediction of cognitive assessment, we denote the neuroimaging data for the $i$-th sample as $x_i$ and the cognitive measure for the $i$-th sample as $y_i$. Suppose that training samples $\mathbf{z} = \{(x_i, y_i)\}_{i=1}^n \subset \mathcal{X} \times \mathcal{Y}$ are generated independently by:

$$Y = f^*(X) + \varepsilon, \tag{1}$$

where $\text{mode}(\varepsilon|X = x) = \arg\max_t p_{\varepsilon|X}(t|X = x) = 0$ for any $x \in \mathcal{X}$. Here, $p_{\varepsilon|X}$, as the conditional density of $\varepsilon$ conditioned on $X$, is well defined. Then, the target function of modal regression can be written as:

$$f^*(x) = \text{mode}(Y|X = x) = \arg\max_t p_{Y|X}(t|X = x), \forall x \in \mathcal{X}. \tag{2}$$

To assure $f^*$ is well defined on $\mathcal{X}$, we require that the existence and uniqueness of $p_{Y|X}(t|X = x)$ for any given $x \in \mathcal{X}$. The relationship (2) means $f^*$ is the maximum of the conditional density $p_{Y|X}$, and also equals to maximize the joint density $p_{X,Y}$ [4, 29, 28]. Here, we formulate the modal regression following the dimension-insensitive statistical learning framework [7].

For feasibility, we denote $\rho$ on $\mathcal{X} \times \mathcal{Y}$ as the intrinsic distribution for data generated by (1), and denote $\rho_{\mathcal{X}}$ as the corresponding marginal distribution on $\mathcal{X}$. It has been proved in Theorem 3 [6] that $f^*$ is the maximizer of

$$\mathcal{R}(f) = \int_{\mathcal{X}} p_{Y|X}(f(x)|X = x)d\rho_{\mathcal{X}}(x) \tag{3}$$

over all measurable function. Hence, we can adopt $\mathcal{R}(f)$ as the evaluation measure of modal regression estimator $f : \mathcal{X} \to \mathbb{R}$. However, we can not get the estimator directly by maximizing this criterion since $p_{Y|X}$ and $\rho_{\mathcal{X}}$ are unknown. Recently, Theorem 5.1 in [6] shows $\mathcal{R}(f) = p_{\varepsilon_f}(0)$, where $p_{\varepsilon_f}$ is the density function of random variable $\varepsilon_f = Y - f(X)$. Then, the problem of maximizing $\mathcal{R}(f)$ over some hypothesis spaces can be transformed to maximize the density of $\varepsilon_f$ at 0. This density $p_{\varepsilon_f}$ can be estimated by nonparametric kernel density estimation.

For a kernel $K_\sigma : \mathbb{R} \times \mathbb{R} \to \mathbb{R}_+$, we denote its representing function $\phi(\frac{u-u'}{\sigma}) = K_\sigma(u, u')$, which usually satisfies $\phi(u) = \phi(-u), \phi(u) \leq \phi(0)$ for any $u \in \mathbb{R}$ and $\int_{\mathbb{R}} \phi(u)du = 1$. Typical examples of kernel include Gaussian kernel, Epanechnikov kernel, quadratic kernel, triwight kernel, and sigmoid function. The empirical estimation of $\mathcal{R}(f)$ (also $p_{\epsilon_f}(0)$) can be obtained by kernel density estimation, which is defined as:

$$\mathcal{R}_{\mathbf{z}}^\sigma(f) = \frac{1}{n\sigma} \sum_{i=1}^n K_\sigma(y_i - f(x_i), 0) = \frac{1}{n\sigma} \sum_{i=1}^n \phi(\frac{y_i - f(x_i)}{\sigma}).$$

Hence, the approximation of $f^*$ can be found by learning algorithms associated with $\mathcal{R}_{\mathbf{z}}^\sigma(f)$. In theory, for any $f : \mathcal{X} \to \mathbb{R}$, the expectation version of $\mathcal{R}_{\mathbf{z}}^\sigma(f)$ is:

$$\mathcal{R}^\sigma(f) = \frac{1}{\sigma} \int_{\mathcal{X} \times \mathcal{Y}} \phi(\frac{y - f(x)}{\sigma})d\rho(x, y).$$

In particular, there holds $\mathcal{R}(f) - \mathcal{R}^\sigma(f) \to 0$ as $\sigma \to 0$ [6].

## 2.2 Modal regression with coefficient-based regularization

In this paper, we assume that $f^*(x) = \text{mode}(Y|X = x) = w_*^T x$ for some $w_* \in \mathbb{R}^p$. Following the ideas of ridge regression and Lasso [22], we consider the robust linear estimator for learning the conditional mode function.

Let $\mathcal{F}$ be a linear hypothesis space defined by:

$$\mathcal{F} = \{f(x) = w^T x : w = (w_1, ..., w_p) \in \mathbb{R}^p, x \in \mathcal{X}\}.$$

For any given positive tuning parameters $\{\tau_j\}_{j=1}^p$, we denote:

$$\Omega(f) = \inf \Big\{ \sum_{j=1}^p \tau_j |w_j|^q : f(x) = w^T x, q \in [1, 2] \Big\}.$$

Given training set $\mathbf{z}$, the regularized modal regression (RMR) can be formulated as below:

$$f_{\mathbf{z}} = \arg\max_{f \in \mathcal{F}} \Big\{ \mathcal{R}_{\mathbf{z}}^\sigma(f) - \lambda\Omega(f) \Big\}, \tag{4}$$

where regularization parameter $\lambda > 0$ is used to balance the modal regression measure and hypothesis space complexity. It is easy to deduce that $f_{\mathbf{z}}(x) = w_{\mathbf{z}}^T x$ with

$$w_{\mathbf{z}} = \arg\max_{w \in \mathbb{R}^p} \Big\{ \frac{1}{n\sigma} \sum_{i=1}^n \phi(\frac{y_i - w^T x_i}{\sigma}) - \lambda \sum_{j=1}^p \tau_j |w_j|^q \Big\}. \tag{5}$$

When $\tau_j \equiv 1$ for $1 \leq j \leq p$ and $q = 1$, (5) can be considered as an natural extension of Lasso in [22] from learning the conditional mean function to estimating the conditional mode function. When $\tau_j \equiv 1$ for $1 \leq j \leq p$ and $q = 2$, (5) also can be regarded as the corresponding version of ridge regression by replacing the MSE criterion with modal regression criterion. In particular, when $K_\sigma$ is Gaussian kernel and $\tau_j \equiv 1$ for $1 \leq j \leq p$, (5) can be rewritten as:

$$w_{\mathbf{z}} = \arg\max_{w \in \mathbb{R}^p} \Big\{ \frac{1}{n\sigma} \sum_{i=1}^n \exp\Big\{ \frac{(y_i - w^T x_i)^2}{\sigma^2} \Big\} - \lambda\|w\|_q^q \Big\},$$

which is equivalent to correntropy regression under maximum correntropy criterion [19, 9, 7].

## 2.3 Optimization algorithm

We employ the half-quadratic (HQ) theory [18] in the optimization. For a convex problem $\min_s u(s)$, it is equivalent to solve the following half-quadratic reformulation:

$$\min_{s,t} Q(s,t) + v(t),$$

where $Q(s,t)$ is quadratic for any $t \in \mathbb{R}$ and $v : \mathbb{R} \to \mathbb{R}$ satisfies:

$$u(s) = \min_t Q(s,t) + v(t), \forall s \in \mathbb{R}.$$

Such a dual potential function $v$ can be determined via convex conjugacy as shown below.

According to the convex optimization theory [20], for a closed convex function $f(a)$, there exists a convex function $g(b)$, such that:

$$f(a) = \max_b(ab - g(b)),$$

where $g$ is the conjugate of $f$, *i.e.*, $g = f^\star$. Symmetrically, it is easy to prove $f = g^\star$.

**Theorem 1** *For a closed convex function $f(a) = \max_b(ab - g(b))$, we have $\arg\max_b(ab - g(b)) = f'(a)$ for any $a \in \mathbb{R}$.*

When $K_\sigma$ is Gaussian kernel, the optimization steps can be found in [9]. Here we take Epanechnikov kernel (*a.k.a.,* parabolic kernel) as an example to show the optimization of Problem (5) via HQ theory. The kernel-induced representing function of Epanechnikov kernel is $\phi(e) = \frac{3}{4}(1 - e^2)\mathbb{1}_{[|e|\leq 1]}$.

Define a closed convex function $f$ as:

$$f(a) = \begin{cases} \frac{3}{4}(1 - a), & 0 \leq a \leq 1 \\ 0, & a \geq 1. \end{cases}$$

There exists a convex function $g$ such that $f(a) = \max_b(ab - g(b))$ and $\phi(e) = f(e^2) = \max_b(e^2 b - g(b))$. Thus, when $\tau_j \equiv 1$ for $1 \leq j \leq p$, the optimization problem (5) can be rewritten as:

$$\max_{w\in\mathbb{R}^p, b\in\mathbb{R}^n} \left\{ \frac{1}{n\sigma} \sum_{i=1}^n \left( b_i (\frac{y_i - w^T x_i}{\sigma})^2 - g(b_i) \right) - \lambda \sum_{j=1}^p \tau_j |w_j|^q \right\}. \tag{6}$$

Problem (6) can be easily optimized via alternating optimization algorithm. Note that according to Theorem 1, when $w$ is fixed, $b$ can be updated as $b_i = f'((\frac{y_i - w^T x_i}{\sigma})^2) = -\frac{3}{4}\mathbb{1}_{[|\frac{y_i - w^T x_i}{\sigma}|\leq 1]}$ for $i = 1, 2, \ldots, n$. *For the space limitation, we provide the proof of Theorem 1 and the optimization steps of RMR in the supplementary material.*

## 3 Learning Theory Analysis

This section presents the theoretical foundations of RMR from approximation ability, variable sparsity, and algorithmic robustness. *Detail proofs of these results can be found in the supplementary material.*

### 3.1 Approximation ability analysis

Besides the linear requirement for the conditional mode function, we also need some basic conditions on the kernel-induced representing function $\phi$ [6, 28].

**Assumption 1** *The representing function $\phi$ satisfies the following conditions: 1) $\forall u \in \mathbb{R}, \phi(u) \leq \phi(0) < \infty$, 2) $\phi$ is Lipschitz continuous with constant $L_\phi$, 3) $\int_\mathbb{R} \phi(u)du = 1$ and $\int_\mathbb{R} u^2\phi(u)du < \infty$.*

It is easy to verify that most of kernels used for density estimation satisfy the above conditions, *e.g.*, Gaussian kernel, Epanechnikov kernel, quadratic kernel, etc. Since RMR is associated with $\mathcal{R}_z^\sigma(f)$, we need to establish quantitative relationship between $\mathcal{R}^\sigma(f)$ and $\mathcal{R}(f)$. Recently, the modal regression calibration has been illustrated in Theorem 10 [6] under the following restrictions on the conditional density $p_{\varepsilon|X}$.

**Assumption 2** *The conditional density $p_{\varepsilon|X}$ is second-order continuously differentiable and uniform bounded.*

Now, we present the approximation bound on $\mathcal{R}(f^*) - \mathcal{R}(f_{\mathbf{z}})$.

**Theorem 2** *Let $\|x\|_{\frac{q}{q-1}} \leq a$ for $q \in (1,2]$ for any $x \in \mathcal{X}$ and $f^* \in \mathcal{F}$. Under Assumptions 1-2, for $q \in (1,2]$, by taking $\lambda = \sigma^2 = O(n^{-\frac{q}{4q+3}})$, we have:*

$$\mathcal{R}(f^*) - \mathcal{R}(f_{\mathbf{z}}) \leq C \log(4/\delta) n^{-\frac{q}{4q+3}}$$

*with confidence at least $1 - \delta$. In particular, for $q = 1$ and $\|x\|_{\infty} \leq a$, choosing $\lambda = \sigma^2 = (\frac{\ln p}{n})^{\frac{1}{7}}$, we have:*

$$\mathcal{R}(f^*) - \mathcal{R}(f_{\mathbf{z}}) \leq C \log(4/\delta) \left(\frac{\ln p}{n}\right)^{\frac{1}{7}}$$

*with confidence at least $1 - \delta$, Here $C_1, C_2$ is a constant independent of $n, \delta$.*

Theorem 2 shows that the excess risk of $\mathcal{R}(f^*) - \mathcal{R}(f_{\mathbf{z}}) \to 0$ with the polynomial decay and the estimation consistency is guaranteed as $n \to \infty$. Moreover, under Assumption 3 in [6], we can derive that $f_{\mathbf{z}}$ tends to $f^*$ with approximation order $O(n^{-\frac{q}{4q+3}})$ for $q \in (1,2]$ and $O(\frac{\ln p}{n})^{\frac{1}{7}})$ for $q = 1$. Although approximation analysis has been provided for modal regression in [6, 28], both of them are limited to the empirical risk minimization. This is different from our result for regularized modal regression under structural risk minimization.

## 3.2 Sparsity analysis

To characterize the variable selection ability of RMR, we first present the properties for nonzero component of $w_{\mathbf{z}}$.

**Theorem 3** *Assume that $\phi$ is differentiable for any $t \in \mathbb{R}$. For $j \in \{1, 2, ..., p\}$ satisfying $w_{\mathbf{z}j} \neq 0$, there holds:*

$$\left| \frac{1}{n\sigma^2} \sum_{i=1}^{n} \phi'\left(\frac{y_i - f_{\mathbf{z}}(x_i)}{\sigma}\right) x_{ij} \right| = \frac{p\lambda\tau_j |w_{\mathbf{z}j}|^{p-1}}{2}.$$

Observe that the condition on $\phi$ holds true for Gaussian kernel, sigmoid function, and logistic function. Theorem 3 demonstrates the necessary condition for the non-zero $w_{\mathbf{z}j}$. Without loss of generality, we set $S_0 = \{1, 2, ..., p_0\}$ as the index set of truly informative variables and denote $S_{\mathbf{z}} = \{j : w_{\mathbf{z}j} \neq 0\}$ as the set of identified informative variables by RMR in (4).

**Theorem 4** *Assume that $\|x\|_{\infty} \leq a$ for any $x \in \mathcal{X}$ and $\lambda\tau_j \geq \|\phi'\|_{\infty}\sigma$ for any $j > p_0$. Then, for RMR (4) with $q = 1$, there holds $S_{\mathbf{z}} \subset S_0$ for all $\mathbf{z} \in (\mathcal{X} \times \mathcal{Y})^n$.*

Theorem 4 assures that RMR has the capacity to identify the truly informative variable in theory. Combining Theorem 4 and Theorem 2, we provide the asymptotic theory of RMR on estimation and model selection.

## 3.3 Robustness analysis

To quantify the robustness of RMR, we calculate its finite sample breakdown point, which reflects the largest amount of contamination points that an estimator can tolerate before returning arbitrary values [11, 12]. Recently, this index has been used to investigate the robustness of modal linear regression [28] and kernel-based modal regression [6].

Recall that the derived weight $w_{\mathbf{z}}$ defined in (5) is dependent on any given sampling set $\mathbf{z} = \{(x_i, y_i)\}_{i=1}^{n}$. By adding $m$ arbitrary points $\mathbf{z}' = \{(x_{n+j}, y_{n+j})\}_{j=1}^{m} \subset \mathcal{X} \times \mathcal{Y}$, we obtain the corrupted sample set $\mathbf{z} \cup \mathbf{z}'$. For given $\lambda, \sigma, \{\tau_j\}_{j=1}^{p}$, we denote $w_{\mathbf{z} \cup \mathbf{z}'}$ be the maximizer of (5). Then, the finite sample breakdown point of $w_{\mathbf{z}}$ is defined as:

$$\epsilon(w_{\mathbf{z}}) = \min_{1 \leq m \leq n} \left\{ \frac{m}{n+m} : \sup_{\mathbf{z}'} \|w_{\mathbf{z} \cup \mathbf{z}'}\|_2 = \infty \right\}.$$

**Theorem 5** *Assume that $\phi(u) = \phi(-u)$ and $\phi(t) \to 0$ as $t \to \infty$. For given $\lambda, \sigma, \{\tau_j\}_{j=1}^p$, we denote:*

$$M = \frac{1}{\phi(0)} \sum_{i=1}^n \phi(\frac{\tilde{y}_i - f_{\mathbf{z}}(x_i)}{\sigma}) - \lambda\sigma(\phi(0))^{-1}\Omega(f_{\mathbf{z}}).$$

*Then the finite sample breakdown point of $w_{\mathbf{z}}$ in (5) is $\epsilon(w_{\mathbf{z}}) = \frac{m^*}{n+m^*}$, where $m^* \geq \lceil M \rceil$ and $\lceil M \rceil$ is the smallest integer not less than $M$.*

From Theorem 5, we know that the finite breakdown point of RMR depends on $\phi, \sigma$, and the sample configuration, which is similar with re-descending M-estimator and recent analysis for modal linear regression in [28]. As illustrated in [11, 12], the finite sample breakdown point is high when the bandwidth $\sigma$ only depends on the training samples. Hence, RMR can achieve satisfactory robustness when $\lambda, \tau_j$ are chosen properly and $\sigma$ is determined by data-driven techniques.

## 4 Experimental Analysis

In this section, we conduct experiments on both toy data, benchmark data as well as the ADNI cohort data to evaluate our RMR model. We compare several regression methods in the experiments, including: **LSR** (traditional mean regression based on the least square estimator), **LSR-L2** (LSR with squared $\ell_2$-norm regularization, *i.e.,* ridge regression) **LSR-L1** (LSR with $\ell_1$-norm regularization), **MedianR** (median regression), **HuberR** (regression with huber loss), **RMR-L2** (RMR with squared $\ell_2$-norm regularization), and **RMR-L1** (RMR with $\ell_1$-norm regularization).

For evaluation, we calculate root mean square error (RMSE) between the predicted value and ground truth in out-of-sample prediction. The RMSE value is normalized via Frobenius norm of the ground truth matrix. We employ 2-fold cross validation and report the average performance for each method. For each method, we set the hyper-parameter of the regularization term in the range of $\{10^{-4}, 10^{-3.5}, \ldots, 10^4\}$. We tune the hyper-parameters via 2-fold cross validation on the training data and report the best parameter *w.r.t.* RMSE of each method. For RMR methods, we adopt the Epanechnikov kernel and set the bandwidth as $\sigma = \max(|y - w^T x|)$.

### 4.1 Performance comparison on toy data

Following the design in [28], we generate the toy data by sampling *i.i.d.* from the model: $Y = -2 + 3X + \tau(X)\epsilon$, where $X \sim \mathcal{U}(0,1)$, $\sigma(X) = 1 + 2X$ and $\epsilon \sim 0.5\mathcal{N}(-2, 3^2) + 0.5\mathcal{N}(2, 1^2)$. We can derive that $E(\epsilon) = 0$, $\text{Mode}(\epsilon) = 1.94$ and $\text{Median}(\epsilon) = 1$, hence the conditional mean regression function of the toy data is $E(Y|X) = -2 + 3X$, the conditional median function is $\text{Median}(Y|X) = 1 + 5X$, while the conditional mode is $\text{Mode}(Y|X) = -0.06 + 6.88X$.

We consider three different number of samples: 100,200,500, and repeat the experiments 100 times for each setting. We present the RMSE in Table 1, which shows that RMR models get lower RMSE values than all comparing methods. It indicates that RMR models make better estimation of the output when the noise in data is skewed and relatively heavy-tailed. Moreover, we compare the coverage probabilities for prediction intervals centered around the predicted value from each method. We set the length of coverage intervals to be $\{0.1\nu, 0.2\nu, 0.3\nu\}$ respectively with $\nu = 3$ being the approximate standard error of $\epsilon$. From Table 2 we can find that RMR models provide larger coverage probabilities than the counterparts.

### 4.2 Performance comparison on benchmark data

Here we present the comparison results on six benchmark datasets from UCI repository [15] and StatLib[2], which include: slumptest, forestfire, bolts, cloud, kidney, and lupus. We summarize the results in Table 3. From the comparison we notice that RMR models tend to perform better on all datasets. Also, RMR-L1 obtains lower RMSE value since the RMR-L1 model is more robust with the $\ell_1$-norm regularization term.

Table 1: Average RMSE and standard deviation with different number (n) of toy samples.

|  | n=100 | n=200 | n=500 |
|---|---|---|---|
| **LSR** | 0.9687±0.0699 | 0.9477±0.0294 | 0.9495±0.0114 |
| **LSR-L2** | 0.9671±0.0685 | 0.9469±0.0284 | 0.9495±0.0114 |
| **LSR-L1** | 0.9672±0.0685 | 0.9473±0.0288 | 0.9495±0.0114 |
| **MedianR** | 0.9944±0.0806 | 0.9568±0.0350 | 0.9542±0.0120 |
| **HuberR** | 0.9725±0.0681 | 0.9485±0.0296 | 0.9502±0.0116 |
| **RMR-L2** | 0.9663±0.0683 | 0.9466±0.0282 | 0.9493±0.0114 |
| **RMR-L1** | **0.9662±0.0679** | **0.9465±0.0281** | **0.9492±0.0114** |

Table 2: Average coverage possibilities and standard deviation on toy data.

|  |  | n=100 | n=200 | n=500 |
|---|---|---|---|---|
| $0.1\nu$ | **LSR** | 0.0730±0.0247 | 0.0702±0.0166 | 0.0702±0.0106 |
|  | **LSR-L2** | 0.0753±0.0247 | 0.0731±0.0155 | 0.0709±0.0108 |
|  | **LSR-L1** | 0.0747±0.0246 | 0.0719±0.0161 | 0.0706±0.0106 |
|  | **MedianR** | 0.0563±0.0255 | 0.0626±0.0124 | 0.0654±0.0097 |
|  | **HuberR** | 0.0710±0.0258 | 0.0698±0.0160 | 0.0694±0.0101 |
|  | **RMR-L2** | **0.0760±0.0254** | 0.0740±0.0161 | 0.0719±0.0111 |
|  | **RMR-L1** | **0.0760±0.0255** | **0.0742±0.0156** | **0.0720±0.0111** |
| $0.2\nu$ | **LSR** | 0.1313±0.0338 | 0.1450±0.0255 | 0.1430±0.0193 |
|  | **LSR-L2** | 0.1337±0.0334 | 0.1461±0.0251 | 0.1429±0.0196 |
|  | **LSR-L1** | 0.1337±0.0337 | 0.1458±0.0258 | 0.1430±0.0193 |
|  | **MedianR** | 0.1087±0.0351 | 0.1331±0.0239 | 0.1377±0.0182 |
|  | **HuberR** | 0.1237±0.0347 | 0.1442±0.0257 | 0.1421±0.0188 |
|  | **RMR-L2** | 0.1340±0.0336 | 0.1477±0.0256 | **0.1441±0.0199** |
|  | **RMR-L1** | **0.1343±0.0340** | **0.1481±0.0247** | **0.1441±0.0198** |
| $0.3\nu$ | **LSR** | 0.1923±0.0402 | 0.2142±0.0342 | 0.2150±0.0229 |
|  | **LSR-L2** | 0.1940±0.0415 | 0.2165±0.0331 | 0.2156±0.0222 |
|  | **LSR-L1** | 0.1940±0.0415 | 0.2153±0.0334 | 0.2153±0.0226 |
|  | **MedianR** | 0.1750±0.0414 | 0.2031±0.0299 | 0.2095±0.0233 |
|  | **HuberR** | 0.1873±0.0389 | 0.2132±0.0333 | 0.2144±0.0224 |
|  | **RMR-L2** | 0.1943±0.0420 | **0.2179±0.0327** | **0.2168±0.0220** |
|  | **RMR-L1** | **0.1950±0.0406** | 0.2177±0.0323 | 0.2167±0.0219 |

Table 3: Average RMSE and standard deviation on benchmark data.

|  | slumptest | forestfire | bolts | cloud | kidney | lupus |
|---|---|---|---|---|---|---|
| **LSR** | 0.2689±0.0295 | 0.9986±0.0874 | 0.4865±0.0607 | 0.6178±0.0190 | 0.5077±0.0264 | 0.8646±0.3703 |
| **LSR-L2** | 0.2616±0.0266 | 0.9822±0.0064 | 0.4687±0.0137 | 0.5782±0.0029 | 0.5106±0.0219 | 0.8338±0.3282 |
| **LSR-L1** | 0.2571±0.0277 | 0.9822±0.0079 | 0.4713±0.0172 | 0.5802±0.0043 | 0.5196±0.0089 | 0.8408±0.3366 |
| **MedianR** | 0.2810±0.0024 | 0.9964±0.0050 | 0.4436±0.0232 | 0.6457±0.0301 | 0.5432±0.0160 | 1.2274±0.6979 |
| **HuberR** | 0.2669±0.0268 | 0.9874±0.0299 | 0.4841±0.0661 | 0.6178±0.0190 | 0.5447±0.0270 | 0.9198±0.4226 |
| **RMR-L2** | 0.2538±0.0185 | 0.9817±0.0093 | 0.4782±0.0107 | 0.5702±0.0131 | **0.4871±0.0578** | 0.8071±0.3053 |
| **RMR-L1** | **0.2517±0.0240** | **0.9802±0.0198** | **0.3298±0.1313** | **0.5663±0.0305** | 0.4989±0.0398 | **0.7885±0.2910** |

Table 4: Average RMSE and standard deviation on the ADNI data.

|  | **Fluency** | **ADAS** | **TRAILS** |
|---|---|---|---|
| **LSR** | 0.3856±0.0034 | 0.4397±0.0112 | 0.6798±0.0538 |
| **LSR-L2** | 0.3269±0.0069 | 0.4116±0.0208 | 0.5443±0.0127 |
| **LSR-L1** | 0.3295±0.0035 | 0.4121±0.0100 | 0.5476±0.0115 |
| **MedianR** | 0.4164±0.0291 | 0.4700±0.0151 | 0.6702±0.1184 |
| **HuberR** | 0.3856±0.0034 | 0.4383±0.0133 | 0.6621±0.0789 |
| **RMR-L2** | **0.3256±0.0049** | 0.4105±0.0216 | **0.5342±0.0186** |
| **RMR-L1** | 0.3269±0.0057 | **0.4029±0.0234** | 0.5423±0.0123 |

### 4.3 Performance comparison on the ADNI cohort data

Now we look into a practical problem in Alzheimer's disease, *i.e.,* prediction of cognitive scores via neuroimaging features. Data used in this article were obtained from the ADNI database (`adni.loni.usc.edu`). We extract 93 regions of interest (ROIs) as neuroimaging features and use cognitive scores from three tests: Fluency Test, Alzheimer's Disease Assessment Scale (ADAS) and Trail making test (TRAILS). 795 sample subjects were involved in our study, including 180 AD samples, 390 MCI samples and 225 normal control (NC) samples. *Detailed data description can be found in the supplementary material.*

Our goal is to construct an appropriate model to predict cognitive performance given neuroimaging data. Meanwhile, we expect the model to illustrate the importance of different features in the prediction, which is fundamental to understanding the role of each imaging marker in the study of AD. From Table 4, we find that RMR models always perform equal or better than the comparing methods, which verifies that RMR is more appropriate to learn the association between neuroimaging markers and cognitive performance. We can notice that RMR-L2 always performs better than LSR-L2, and RMR-L1 outperforms LSR-L1. This is because the symmetric noise assumption in least square models may not be guaranteed on the ADNI cohort. Compared with HuberR, our RMR model is shown to be less sensitive to outliers. Moreover, from the comparison between MedianR and RMR models, we can infer that conditional mode is more suitable than conditional median for the prediction of cognitive scores.

RMR-L1 imposes sparse constraints on the learnt weight matrix, which naturally achieves the goal of feature selection in the association study. Here we take TRAILS cognitive assessment as an example and look into the important neuroimaging features in the prediction. From the heat map and brain map in Fig. 1 and 2, we obtain several interesting findings. In the prediction, temporal lobe white matter has been picked out as a predominant feature. [10, 2] reported decreased fractional anisotropy (FA) and increased radial diffusivity (DR) in the white matter of the temporal lobe among AD and Mild Cognitive Impairment (MCI) subjects. [10] also revealed the correlation between temporal lobe FA and episodic memory, which may account for the influence of temporal lobe to TMT results. Besides, there is evidence in [17] supporting the association between left temporal lobe and the working memory component involving letters and numbers in TMT. Moreover, angular gyrus indicates high correlation with TRAILS scores in our analysis. Previous research has revealed that angular gyrus share many clinical features with AD. [14] presented structural MRI findings showing more left anular gyrus in MCI converters than non-converters, which pointed out the role of atrophy of structures like angular gyrus in the progression of dementia. [3] showed evidence for the role of angular gyrus in orienting spatial attention, which serves as a key factor in TMT results. The replication of these results supports the effectiveness of our model.

## 5 Conclusion

This paper proposes a new regularized modal regression and establishes its theoretical foundations on approximation ability, sparsity, and robustness. These characterizations fill in the theoretical gaps for modal regression under Tikhonov regularization. Empirical results verify the competitive performance of the proposed approach on simulated data, benchmark data and real biomedical data.

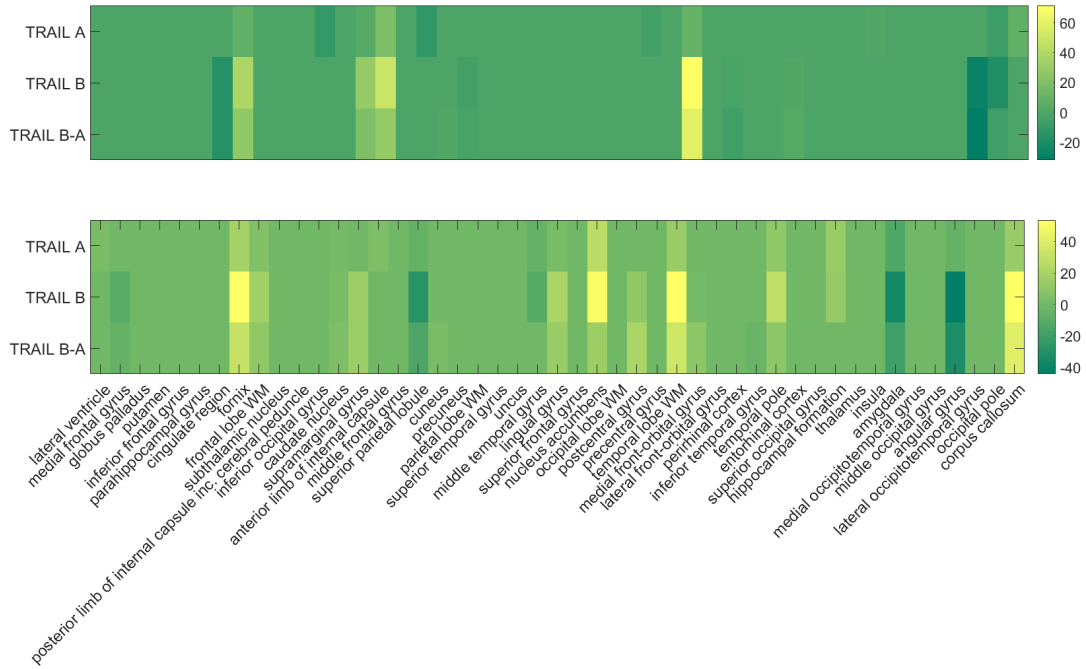

Figure 1: Heatmap showing the weights of each neuroimaging feature via RMR-L1 model for the prediction of TRAILS cognitive measures. We draw two matrices, where the upper figure is for the left hemisphere and the lower figure for the right hemisphere. Imaging markers (columns) with larger weights indicate higher correlation with corresponding cognitive measure in the prediction.

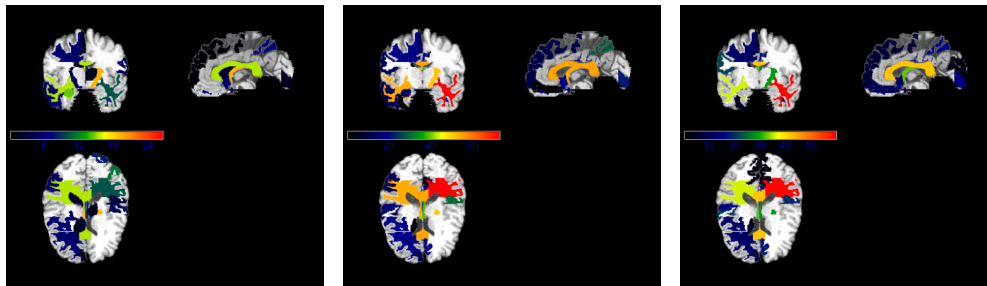

Figure 2: Cortical maps of ROIs identified in RMR-L1 model for the prediction of TRAILS cognitive measures. The brain maps show one slice of multi-view. The three maps correspond to three different cognitive measures in TRAILS cognitive test, respectively.

With the sparsity property of our model, we identified several biological meaningful neuroimaging markers, showing the potential to enhance the understanding of onset and progression of AD.

# Acknowledgments

This work was partially supported by U.S. NSF-IIS 1302675, NSF-IIS 1344152, NSF-DBI 1356628, NSF-IIS 1619308, NSF-IIS 1633753, NIH AG049371. Hong Chen was partially supported by National Natural Science Foundation of China (NSFC) 11671161. We are grateful to the anonymous NIPS reviewers for the insightful comments.

## Footnotes

*X. Wang and H. Chen made equal contributions to this paper. H. Huang is the corresponding author.

[2]http://lib.stat.cmu.edu/datasets/

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
