[Supplementary Material]

# Supplementary Material to "Regularized Modal Regression with Applications in Cognitive Impairment Prediction"

**Xiaoqian Wang**[1]**, Hong Chen**[1]**, Weidong Cai**[2]**, Dinggang Shen**[3]**, Heng Huang**[1]*

[1] Department of Electrical and Computer Engineering, University of Pittsburgh, USA
[2]School of Information Technologies, University of Sydney, Australia
[3] Department of Radiology and BRIC, University of North Carolina at Chapel Hill, USA
xqwang1991@gmail.com,chenh@mail.hzau.edu.cn
tom.cai@sydney.edu.au,dinggang_shen@med.unc.edu,heng.huang@pitt.edu

## 1 Optimization steps of RMR

We summarize the optimization steps of Regularized Modal Regression (RMR) in Algorithm 1.

## 2 Proof of Theorem 1

**Proof:** For any $a_0 \in \mathbb{R}$, suppose $b_0 = \arg\max_b (a_0 b - g(b))$, then we have $f(a_0) = a_0 b_0 - g(b_0)$, thus $g(b_0) = a_0 b_0 - f(a_0)$. Since $g = f^\star$, we obtain that $g(b_0) = \max_a (ab_0 - f(a))$. Taking derivative of the maximum function *w.r.t.* $a$, we find that the maximum is reached when $f'(a) = b_0$. Since $a_0 = \arg\max_a (ab_0 - f(a))$, we get $f'(a_0) = b_0$. This completes the proof.

## 3 Proof of Theorem 2

The approximation bound in Theorem 2 is estimated by error analysis technique associated with Rademacher complexity.

We introduce a data-free function $f_\lambda = w_\lambda^T x$ as the stepping-stone for decomposing $\mathcal{R}(f^*) - \mathcal{R}(f_\mathbf{z})$, where

$$w_\lambda = \underset{w \in \mathbb{R}^p}{\arg\max} \left\{ \frac{1}{\sigma} \int_{\mathcal{X} \times \mathcal{Y}} \phi\left(\frac{y - w^T x}{\sigma}\right) d\rho(x, y) - \lambda \sum_{j=1}^{p} \tau_j |w_j|^p \right\}.$$

Here, similar with RMR in Section 2, $\lambda > 0$ is a regularized parameter and $\{\tau_j\}$ are weights for different input components.

From the viewpoint of function learning, we have

$$f_\lambda = \underset{f \in \mathcal{F}}{\arg\max} \left\{ \mathcal{R}^\sigma(f) - \lambda \Omega(f) \right\}.$$

Now we present the decomposition for $\mathcal{R}(f^*) - \mathcal{R}(f_\mathbf{z})$.

**Algorithm 1** Half Quadratic Optimization Algorithm for RMR.
---
**Require:** Input data $\mathbf{z} = \{(x_i, y_i)\}_{i=1}^n$, and the choice of kernel-induced representing function $\phi$.
**Ensure:** $w_{\mathbf{z}}$
 1: Define function $f$ such that $f(e^2) = \phi(e)$;
 2: Initialize bandwidth $\sigma$;
 3: Initialize $w$ randomly;
 4: **while** not converge **do**
 5:   Update $b_i$ such that $b_i = f'((\frac{y_i - w^T x_i}{\sigma})^2)$, for $i = 1, 2, \ldots, n$;
 6:   Update $w$ such that $w = \underset{w \in \mathbb{R}^p}{\arg\max} \frac{1}{n\sigma} \sum_{i=1}^n \left( b_i(\frac{y_i - w^T x_i}{\sigma})^2 - g(b_i) \right) - \lambda \sum_{j=1}^p \tau_j |w_j|^q$;
 7:   Update $\sigma$;
 8: **end while**
 9: $w_{\mathbf{z}} = w$.
---

**Proposition 1** *Let $f^* \in \mathcal{F}$. Under Assumptions 1 and 2, there holds*

$$
\begin{aligned}
&\mathcal{R}(f^*) - \mathcal{R}(f_{\mathbf{z}}) \\
\leq\ &\mathcal{R}_{\mathbf{z}}^\sigma(f_{\mathbf{z}}) - \mathcal{R}^\sigma(f_{\mathbf{z}}) + \mathcal{R}^\sigma(f_\lambda) - \mathcal{R}_{\mathbf{z}}^\sigma(f_\lambda) + \sigma^2 \|p_{\varepsilon|X}''\|_\infty \int_{\mathbb{R}} u^2 \phi(u) du + \lambda \Omega(f^*).
\end{aligned}
$$

**Proof:** According to Theorem 10 in [5], we know that

$$
\mathcal{R}(f^*) - \mathcal{R}(f_{\mathbf{z}}) \leq \mathcal{R}(f^*) - \mathcal{R}^\sigma(f_{\mathbf{z}}) + \sigma^2 \|p_{\varepsilon|X}''\|_\infty \int_{\mathbb{R}} u^2 \phi(u) du. \tag{1}
$$

Moreover, from the definitions of $f_\lambda$ and $f_{\mathbf{z}}$, we obtain that

$$
\begin{aligned}
\mathcal{R}^\sigma(f^*) - \mathcal{R}^\sigma(f_{\mathbf{z}}) &= \mathcal{R}^\sigma(f^*) - \lambda\Omega(f^*) - \mathcal{R}^\sigma(f_{\mathbf{z}}) + \lambda\Omega(f^*) \\
&\leq \mathcal{R}^\sigma(f_\lambda) - \lambda\Omega(f_\lambda) - \mathcal{R}^\sigma(f_{\mathbf{z}}) + \lambda\Omega(f^*) \\
&\leq \mathcal{R}^\sigma(f_\lambda) - \mathcal{R}_{\mathbf{z}}^\sigma(f_\lambda) + \left\{ \mathcal{R}_{\mathbf{z}}^\sigma(f_\lambda) - \lambda\Omega(f_\lambda) - \left( \mathcal{R}_{\mathbf{z}}^\sigma(f_{\mathbf{z}}) - \lambda\Omega(f_{\mathbf{z}}) \right) \right\} \\
&\quad + \mathcal{R}_{\mathbf{z}}^\sigma(f_{\mathbf{z}}) - \mathcal{R}^\sigma(f_{\mathbf{z}}) + \lambda\Omega(f^*) \\
&\leq \mathcal{R}^\sigma(f_\lambda) - \mathcal{R}_{\mathbf{z}}^\sigma(f_\lambda) + \mathcal{R}_{\mathbf{z}}^\sigma(f_{\mathbf{z}}) - \mathcal{R}^\sigma(f_{\mathbf{z}}) + \lambda\Omega(f^*).
\end{aligned}
$$

Combining the above decomposition with (1), we get the desire result.

The error term $\mathcal{R}^\sigma(f_\lambda) - \mathcal{R}_{\mathbf{z}}^\sigma(f_\lambda)$ can be bounded by the Bernstein inequality [2, 4].

**Lemma 1** *Let $\xi$ be a random variable on a probability space $\mathcal{Z}$ with mean $E\xi$ and variance $\nu$. If $|\xi(z) - E\xi| \leq M_\xi$ for almost all $z \in \mathcal{Z}$, then with confidence at least $1 - \delta$*

$$
E\xi - \frac{1}{m} \sum_{i=1}^m \xi(z_i) \leq \frac{2M_\xi \log(1/\delta)}{3m} + \sqrt{\frac{2\nu^2 \log(2/\delta)}{m}}.
$$

**Proposition 2** *Under Assumption 1, for any $\delta \in (0, 1)$, there holds*

$$
\mathcal{R}^\sigma(f_\lambda) - \mathcal{R}_{\mathbf{z}}^\sigma(f_\lambda) \leq \frac{4\phi(0) \log(2/\delta)}{3n\sigma} + \sigma^{-1}\phi(0)\sqrt{\frac{2\ln(2/\delta)}{n}}
$$

*with confidence at least $1 - \delta$.*

**Proof:** Denote $\xi(x, y) = \sigma^{-1}\phi(\frac{y - f(x)}{\sigma})$ for any $(x, y) \in \mathcal{X} \times \mathcal{Y}$ and $f \in \mathcal{F}$. We can verify that $0 < \xi(x, y) \leq \sigma^{-1}\phi(0)$ and $|\xi - E\xi| \leq \sigma^{-1}\phi(0)$. Then, according to Bernstein inequality in Lemma 1, we have

$$
\mathcal{R}^\sigma(f_\lambda) - \mathcal{R}_{\mathbf{z}}^\sigma(f_\lambda) = E\xi - \frac{1}{n} \sum_{i=1}^n \xi(x_i, y_i) \leq \frac{4\phi(0) \log(2/\delta)}{3n\sigma} + \sigma^{-1}\phi(0)\sqrt{\frac{2\ln(2/\delta)}{n}}
$$

with confidence at least $1 - \delta$. This completes the proof.

Since $f_{\mathbf{z}}$ varies with random drawn samples $\mathbf{z}$, we need to provide the uniform estimation on the error term $\mathcal{R}_{\mathbf{z}}^{\sigma}(f_{\mathbf{z}}) - \mathcal{R}^{\sigma}(f_{\mathbf{z}})$ for all $\mathbf{z} \in (\mathcal{X} \times \mathcal{Y})^n$. Here, we adopt the concentration estimation in [1, 8] to bound this error term, where the capacity of hypothesis space is measured by Rademacher complexity. In particular, Rademacher complexity for linear function classes has been well characterized in [8, 7, 9].

**Definition 1** *Let $\{z_i\}_{i=1}^n \in \mathcal{Z}^n$ be independent samples selected according to $\mu$ and let $\mathcal{G}$ be a class of functions mapping from $\mathcal{Z}$ to $\mathbb{R}$. Define the Rademacher complexity of $\mathcal{G}$ to be*

$$\mathcal{R}_n(\mathcal{G}) = E_\mu E_\epsilon \Big[ \sup_{g \in \mathcal{G}} \frac{1}{n} \sum_{i=1}^n \epsilon_i g(z_i) \Big],$$

*where $\{\epsilon_i\}_{i=1}^n$ are independent random variables uniformly chosen from $\{-1, 1\}$.*

The following concentration inequality has been used extensively for error analysis, see, e.g., [1, 8, 7, 9] .

**Lemma 2** *Assume that loss function $\psi(f, z)$ is $L$ Lipschitz continuous with respect to $f$ and $|\psi(f, z)| \le c$ for any $z \in \mathcal{Z}$ and $f \in \mathcal{G}$. For any $\delta \in (0, 1)$, with confidence $1 - \delta$ there holds*

$$\frac{1}{n} \sum_{i=1}^n \psi(f, z_i) - E\psi(f, z) \le 2L\mathcal{R}_n(\mathcal{G}) + c\sqrt{\frac{\ln(2/\delta)}{2n}}.$$

**Proposition 3** *Assume that $\|x\|_{\frac{q}{q-1}} \le a$ for $q \in (1, 2]$ and any $x \in \mathcal{X}$. Let Assumption 1 be true. For $q \in (1, 2]$ and any $\delta \in (0, 1)$, with confidence at least $1 - \delta$ there holds*

$$\mathcal{R}_{\mathbf{z}}^{\sigma}(f_{\mathbf{z}}) - \mathcal{R}^{\sigma}(f_{\mathbf{z}}) \le 2aL_\phi \Big( \frac{\phi(0)}{n\lambda\sigma^{2q+1}} \Big)^{\frac{1}{q}} + 2\sigma^{-1}\phi(0)\sqrt{\frac{\ln(4/\delta)}{n}}.$$

*For $q = 1$ and $\|x\|_\infty \le a$ for any $x \in \mathcal{X}$, with confidence at least $1 - \delta$, there holds*

$$\mathcal{R}_{\mathbf{z}}^{\sigma}(f_{\mathbf{z}}) - \mathcal{R}^{\sigma}(f_{\mathbf{z}}) \le \frac{2\sqrt{2}a\sigma^{-3}\lambda^{-1}L_\phi\phi(0)(2 + \sqrt{\ln p})}{\sqrt{n}} + 2\sigma^{-1}\phi(0)\sqrt{\frac{\ln(4/\delta)}{n}}.$$

**Proof:** From the definition of $f_{\mathbf{z}}$, we know that

$$\mathcal{R}_{\mathbf{z}}^{\sigma}(f_{\mathbf{z}}) - \lambda\Omega(f_{\mathbf{z}}) \ge \mathcal{R}_{\mathbf{z}}^{\sigma}(0).$$

It means that

$$\Omega(f_{\mathbf{z}}) = \sum_{j=1}^p \tau_j |w_{\mathbf{z},j}|^q \le \lambda^{-1}\sigma^{-1}\phi(0).$$

Then, we can deduce that

$$\|w_{\mathbf{z}}\|_q = \Big( \sum_{j=1}^p |w_{\mathbf{z},j}|^q \Big)^{\frac{1}{q}} \le \Big( \frac{\phi(0)}{\lambda\sigma \min_j \tau_j} \Big)^{\frac{1}{q}}.$$

We first consider the setting of $q \in (1, 2]$. Denote

$$\mathcal{G}_1 = \{f(x) = w^T x : \|w\|_q \le \Big( \frac{\phi(0)}{\lambda\sigma \min_j \tau_j} \Big)^{\frac{1}{q}}, \|x\|_{\frac{q}{q-1}} \le a\}.$$

Notice that $\|w\|_q$ is $q - 1$-strongly convex on $\mathbb{R}^d$ with respect to $\|\cdot\|_q$ [8, 7]. According to Theorem 3 in [8], we have

$$\mathcal{R}_n(\mathcal{G}_1) \le a\Big( \frac{\phi(0)}{\lambda\sigma \min_j \tau_j} \Big)^{\frac{1}{q}} \sqrt{\frac{1}{n(q-1)}}.$$

Let $\psi(f, z) = \sigma^{-1}\phi(\frac{y - f(x)}{\sigma}), f \in \mathcal{G}_1$. For any measurable functions $f_1, f_2$, and $(x, y) \in \mathcal{X} \times \mathcal{Y}$, there holds

$$\Big| \psi(f_1, z) - \psi(f_2, z) \Big| = \Big| \frac{1}{\sigma}\phi(\frac{y - f_1(x)}{\sigma}) - \frac{1}{\sigma}\phi(\frac{y - f_2(x)}{\sigma}) \Big| \le \sigma^{-2}L_\phi |f_1(x) - f_2(x)|.$$

This means $\psi(f,z)$ has Lipschitz constant $\sigma^{-2}L_\phi$ with respect to $f$. From Assumption 1, we know that $\psi(f,z) \le \sigma^{-1}\phi(0)$. Applying Lemma 2 to $\psi(f,z), f \in \mathcal{G}_1$, we obtain that

$$\forall f \in \mathcal{G}_1, \frac{1}{n}\sum_{i=1}^m \psi(f,z_i) - E\psi \le 2aL_\phi(\frac{\phi(0)}{n\lambda\sigma^{2q+1}})^{\frac{1}{q}} + 2\sigma^{-1}\phi(0)\sqrt{\frac{\ln(4/\delta)}{n}}$$

with confidence at least $1-\delta$. This asserts the first statement as $f_{\mathbf{z}} \in \mathcal{G}_1$.

Now we turn to consider the setting of $q = 1$. When $q = 1$, we can deduce that $\Omega(f_\lambda) = \sum_{j=1}^p |w_{\mathbf{z},j}| \le \lambda^{-1}\sigma^{-1}\phi(0)$. Denote

$$\mathcal{G}_2 = \{f(x) = w^T x : \Omega(f_\lambda) \le \lambda^{-1}\sigma^{-1}\phi(0), \|x\|_\infty \le a\}.$$

According to Theorem 2 in [9], we have

$$\mathcal{R}_n(\mathcal{G}_2) = \frac{a\lambda^{-1}\sigma^{-1}\phi(0)(2\sqrt{2}+\sqrt{2\ln p})}{n}.$$

Similar with the proof procedures for $q \in (1,2]$, we have with confidence $1-\delta$

$$\mathcal{R}_{\mathbf{z}}^\sigma(f_{\mathbf{z}}) - \mathcal{R}^\sigma(f_{\mathbf{z}}) \le \frac{2\sqrt{2}a\sigma^{-3}\lambda^{-1}L_\phi\phi(0)(2+\sqrt{\ln p})}{\sqrt{n}} + 2\sigma^{-1}\phi(0)\sqrt{\frac{\ln(4/\delta)}{n}}.$$

by applying Lemma 2 to $\psi(f,z) = \sigma^{-1}\phi(\frac{y-f(x)}{\sigma}), f \in \mathcal{G}_2$. The second statement in Proposition 3 is proved.

It is a position to present the proof of Theorem 2.

**Proof of Theorem 2**: For $q \in (1,2]$, by combining Propositions 1- 3, we get with confidence at least $1 - 2\delta$

$$\mathcal{R}(f^*) - \mathcal{R}(f_{\mathbf{z}}) \le C_1 \log(2/\delta)(\sigma^{-\frac{5}{2}}\lambda^{-\frac{1}{q}}n^{-\frac{2q+1}{q}} + \sigma^2 + \lambda), \tag{2}$$

where $C_1 > 0$ is a constant depending on $\phi(0), f^*, L_\phi$ and $q$.

Setting $\lambda = \sigma^2 = \sigma^{-\frac{2q+1}{q}}\lambda^{-\frac{1}{q}}n^{-\frac{1}{2}}$, we derive $\sigma = n^{-\frac{q}{8q+6}}$ and $\lambda = n^{-\frac{q}{4q+3}}$. Then, from (2), we get

$$\mathcal{R}(f^*) - \mathcal{R}(f_{\mathbf{z}}) \le 3C_1 \log(4/\delta)n^{-\frac{q}{4q+3}}$$

with confidence at least $1-\delta$. This proves the first statement of Theorem 2.

When $q = 1$, from Propositions 1- 3, we get with confidence at least $1 - 2\delta$

$$\mathcal{R}(f^*) - \mathcal{R}(f_{\mathbf{z}}) \le C_2 \log(2/\delta)(\sigma^{-3}\lambda^{-1}\sqrt{\frac{\ln p}{n}} + \sigma^2 + \lambda), \tag{3}$$

where $C_2 > 0$ is a constant independent of $n, \delta$.

Setting $\lambda = \sigma^2 = \sigma^{-3}\lambda^{-1}n^{-\frac{1}{2}}\sqrt{\ln p}$, we obtain that $\sigma = (\frac{\ln p}{n})^{\frac{1}{14}}$ and $\lambda = (\frac{\ln p}{n})^{\frac{1}{7}}$. Then, from (3), we deduce that

$$\mathcal{R}(f^*) - \mathcal{R}(f_{\mathbf{z}}) \le 3C_2 \log(4/\delta)\Big(\frac{\ln p}{n}\Big)^{\frac{1}{7}}$$

with confidence at least $1-\delta$. This completes the proof.

# 4 Proof of Theorems 3 and 4

The sparsity characterization in Theorems 3 and 4 are obtained in terms of the properties of resulting estimator and analysis procedures in [11, 13, 3, 16]. We first provide the proof of sparsity characterization for non-zero pattern of $w_{\mathbf{z}}$.

**Proof of Theorem 3**: Denote

$$G(w) = \frac{1}{n\sigma}\sum_{i=1}^n \phi(\frac{y_i - w^T x_i}{\sigma}) - \lambda\sum_{j=1}^p \tau_j|w_j|^q, 1 \le q \le 2.$$

Let $I_+ = \{j : w_{\mathbf{z},j} > 0\}$ and $I_- = \{j : w_{\mathbf{z},j} < 0\}$. For $j \in I_+$, there holds

$$\frac{\partial G(w)}{\partial w_j}\Big|_{w=w_\mathbf{z}} = -\frac{1}{n\sigma^2}\sum_{i=1}^{n}\phi'(\frac{y_i - f_\mathbf{z}(x_i)}{\sigma})x_{ij} - \lambda q \tau_j w_{\mathbf{z}j}^{q-1} = 0.$$

This means

$$-\frac{1}{n\sigma^2}\sum_{i=1}^{n}\phi'(\frac{y_i - f_\mathbf{z}(x_i)}{\sigma})x_{ij} = \lambda q \tau_j w_{\mathbf{z}j}^{q-1}. \tag{4}$$

Similarly, for $j \in I_-$, there is

$$\frac{\partial G(w)}{\partial w_j}\Big|_{w=w_\mathbf{z}} = -\frac{1}{n\sigma^2}\sum_{i=1}^{n}\phi'(\frac{y_i - f_\mathbf{z}(x_i)}{\sigma})x_{ij} + \lambda q \tau_j w_{\mathbf{z}j}^{q-1} = 0.$$

Then,

$$-\frac{1}{n\sigma^2}\sum_{i=1}^{n}\phi'(\frac{y_i - f_\mathbf{z}(x_i)}{\sigma})x_{ij} = -\lambda q \tau_j w_{\mathbf{z}j}^{a-1}. \tag{5}$$

Combining (4) and (7), we know that for $j$ satisfying $w_{\mathbf{z},j} \neq 0$ such that

$$-\frac{1}{n\sigma^2}\sum_{i=1}^{n}\phi'(\frac{y_i - f_\mathbf{z}(x_i)}{\sigma})x_{ij} = \lambda q \tau_j |w_{\mathbf{z}j}|^{q-1}.$$

This completes the proof.

The proof of Theorem 4 is inspired from the sparsity analysis in [16].

**Proof of Theorem 4**: Suppose that $w_{\mathbf{z}j} \neq 0$ for some $j > p_0$. Observe that

$$\Big|\frac{1}{n\sigma^2}\sum_{i=1}^{n}\phi'(\frac{y_i - f_\mathbf{z}(x_i)}{\sigma})x_{ij}\Big| \leq \frac{1}{n\sigma^2}\sum_{i=1}^{n}|\phi'(\frac{y_i - f_\mathbf{z}(x_i)}{\sigma})x_{ij}| \leq \frac{\|\phi'\|_\infty\|x\|_\infty}{\sigma^2} \leq \frac{a\|\phi'\|_\infty}{\sigma^2}.$$

This together with Theorem 3 implies that $q\lambda\tau_j|w_{\mathbf{z}j}|^{q-1} \leq a\phi^{-2}\|\phi'\|_\infty$. For $q = 1$, this result contradicts with the parameter condition $\lambda\tau_j > a\phi^{-2}\|\phi'\|_\infty$. Hence, we have $w_{\mathbf{z}j} = 0$ for any $j > p_0$. This proves the assertion of Theorem 4.

## 5 Proof of Theorem 5

The robust result in Theorem 5 is inspired from recent related works in [17, 5]. We use the analysis strategy in [17] to establish the proof of Theorem 5.

**Proof of Theorem 5**: The RMR in Section 2 is equivalent to the following optimization

$$\max\Big\{\sum_{i=1}^{n}\frac{\phi(\frac{y_i - f(x_i)}{\sigma})}{\phi(0)} - n\lambda\sigma(\phi(0))^{-1}\sum_{j=1}^{p}\tau_j|w_j|^p\Big\}.$$

Denote $\phi^*(t) = \phi(t)/\phi(0)$. Then,$\forall t, \phi^*(t) \leq \phi^*(0) = 1$ and $\phi^*(t) \to 0$ as $t \to \infty$. When $m < M$, there exists $m + n\zeta < M$ for some $\zeta > 0$. Let $\phi^*(t) \leq \zeta$ for $|t| \geq c$ and let $w$ be any real vector such that $|y - w^T x| \geq c$ for any $(x, y) \in \mathbf{z}$. Then, we deduce that

$$\sum_{i=1}^{n+m}\phi^*(y_i - w_\mathbf{z}^T x_i) - \lambda\sigma(\phi(0))^{-1}\sum_{j=1}^{p}\tau_j|w_{\mathbf{z}j}|^q \geq M \tag{6}$$

and

$$\begin{aligned}\sum_{i=1}^{n+m}\phi^*(y_i - w^T x_i) - \lambda\sigma(\phi(0))^{-1}\sum_{j=1}^{p}\tau_j|w_j|^q &\leq \sum_{i=n+1}^{n+m}\phi^*(y_i - w^T x_i) + \sum_{i=1}^{n}\phi^*(y_i - w^T x_i)\\ &\leq m + n\zeta.\end{aligned} \tag{7}$$

Combining (4) and (7), we have

$$\sum_{i=1}^{n+m} \phi^*(y_i - w_{\mathbf{z}}^T x_i) - \lambda\sigma(\phi(0))^{-1} \sum_{j=1}^{p} \tau_j |w_{\mathbf{z},j}|^q \geq \sum_{i=1}^{n+m} \phi^*(y_i - w^T x_i) - \lambda\sigma(\phi(0))^{-1} \sum_{j=1}^{p} \tau_j |w_j|^q.$$

According to the definition of $w_{\mathbf{z}\cup\mathbf{z}'}$, one knows that $w_{\mathbf{z}\cup\mathbf{z}'}$ must satisfy $|y - w_{\mathbf{z}\cup\mathbf{z}'}^T x| < c$ for at least one point in $\mathbf{z}$. Hence, $w_{\mathbf{z}\cup\mathbf{z}'}$ is bounded as $m < M$. That is to say the finite sample breakdown point of RMR is larger than $M$.

## 6 Data description

Data used in this article were obtained from the ADNI database (`adni.loni.usc.edu`). Each MRI T1-weighted image was first anterior commissure (AC) posterior commissure (PC) corrected using MIPAV2, intensity inhomogeneity corrected using the N3 algorithm [12], skull stripped [15] with manual editing, and cerebellum-removed [14]. We then used FAST [18] in the FSL package3 to segment the image into gray matter (GM), white matter (WM), and cerebrospinal fluid (CSF), and used HAMMER [10] to register the images to a common space. GM volumes obtained from 93 ROIs defined in [6], normalized by the total intracranial volume, were extracted as features. Cognitive scores were obtained from three independent cognitive assessments including Fluency Test, Alzheimer's Disease Assessment Scale (ADAS) and Trail making test (TRAILS). Details of these cognitive assessments can be found in the ADNI procedure manuals. All participants with no missing baseline MRI measurements and cognitive measures were included in this study. In total, there are 795 sample subjects in our study, including 180 AD samples, and 390 MCI samples and 225 normal control (NC) samples. Five cognitive scores were employed in the experiments, which are: 1) ADAS cognitive score; 2) FLU ANIM and FLU VEG scores from Fluency cognitive assessment; 3) Trails A and Trails B scores from Trail making test.

## Footnotes

*X. Wang and H. Chen made equal contributions to this paper. H. Huang is the corresponding author.