[Reviews · NeurIPS 2017]

Reviewer 1



Summary: In this paper the authors extend the modal regression framework with sparse regularization (combining modal regression and LASSO). They provide learning theory analysis for regularized modal regression (RMR). They evaluate the method on simulated toy data and in addition they apply the method to a clinical problem in Alzheimer’s disease: Predicting cognitive scores from brain imaging data. As a data source they use the publicly available ADNI database. The proposed method is compared to a suitable set of alternative and established method (least squares regression, ridge and lasso, and median regression). Overall the proposed method performs slightly better than competing methods. However, the improvement over related methods such as ridge regression is minuscule, and looking at the accompanying standard deviation, not statistically significant. This may have to do with the task at hand and the method may be more suitable for other problems. Comments: 1. The ADNI databse contains more than 1500 subjects with imaging and scores from cognitive tests, why were only 795 subjects used? 2. Page 7: that is a very strong statement given the observed small differences between methods “The reason that least square models do not work well is that the assumption of symmetric noise may not be guaranteed on the ADNI cohort.” E.g., for “Fluency” ridge regression achieved an RMSE 0.3269+/-0.006 and RMR-L2 0.3276+/-0.0049. 3. Figure 1: the text is too small to read the axis annotation (i.e., feature names). Minor: 1. Page 3 in 2.2: “consider consider” 2. Page 5 in 3.3. “Before return arbitrary” -> “Before returning arbitrary”

Reviewer 2



This manuscript formulates modal regression as a penalized linear regression with a new loss, applied to prediction of Alzheimer's disease status from brain images. The new loss function captures the mode, rather than the mean, of the output variable. Such modal regression has been developped previously, but the manuscript studies it in the context of risk minimization with l1 or l2 regularization, and gives interesting theoretical results in those settings. The theoretical contribution is interesting, as the resulting formulation seems quite easy to work with. In the empirical validation, the difference in RMSE seems non significant. I wonder if this is not bound to happen: the RMR methods do not seek to minimize RMSE. I find the NeuroImaging validation not convincing: although I myself do a lot of neuroimaging data analysis, the sample sizes are small, and it would be interesting to confirm the effects on other data with more samples. Also, it would be interesting to compare to a huber loss, which is also robust. The intuition is that huber would probably be less robust.

Reviewer 3



The authors present a regularized modal regression method. The statistical learning view of this proposed method is studied and the resulting model is applied to Alzheimer's disease studies. There are several presentation and evaluation issues for this work in its current form. Firstly, the authors motivate the paper using Alzheimer's disease studies and argue that modal regression is the way to analyze correlations between several disease markers of the disease. This seems very artificial. The necessity of use conditional mode for regression has nothing specific for the Alzheimer's application. The motivation for RMR makes sense without any AD related context. The authors then argue that the main contribution for the work is designing modal regression for AD (line 52). However, for the rest of the methods the discussion is about statistical learning -- no AD at all. This presentation is dubious. The authors need to decide if the work is statistical learning theory for modal regression OR an application of modal regression for AD (for the latter more applied journals are relevant and not this venue). Moving beyond this overall presentation issue, if we assume that modal regression for AD is the focus of the paper, then the results should show precisely this performance improvement compared to the rest mean-based regression. However, as shown in section 4 this does not seem to be the case. In several cases the performance of LSR is very close to RMR. It should be pointed out that when reporting such sets of error measures, one should perform statistical tests to check if the performance difference is significant i.e., p-value differences between all the entries of Tables. Beyond, the tables the message is not clear in the Figures -- the visualizations need to be improved (or quantified in some sense). This non-appealing improvements of AD data evaluations is all the more reason that the authors should choose 'standard' regression datasets and show the performance gains of RMR explicitly (once this is done, the results of AD can be trusted and any small improvement can be deemed OK).